# Developmental Changes in the Locus of Control in Students Attending Integrated and Non-integrated Classes during Early Adolescence in Poland

**DOI:** 10.3390/bs10040074

**Published:** 2020-04-07

**Authors:** Beata Łubianka, Sara Filipiak, Katarzyna Mariańczyk

**Affiliations:** 1Department of Psychology, Jan Kochanowski University in Kielce, 25-369 Kielce, Poland; 2Institute of Psychology, Maria Curie-Skłodowska University, 20-400 Lublin, Poland; s.filipiak@poczta.umcs.lublin.pl; 3Institute of Psychology, The John Paul II Catholic University of Lublin, 20-950 Lublin, Poland; marianczyk@kul.pl

**Keywords:** locus of control, early adolescence, developmental changes, integrated education

## Abstract

This article reports the results of a longitudinal study on the development of context-specific locus of control related to situations of success and failure in Polish adolescents. The participants were 90 primary school students, including 30 who learned in integrated classrooms and 60 who went to non-integrated classes in schools with and without an inclusive curriculum, located in Lublin, Poland. The students were surveyed during a three-year schooling period (when they were in the sixth, seventh, and eighth grade). The research was carried out in the years 2016–2019. The Locus of Control Questionnaire (LOQ and LOQ-R) by Krasowicz-Kupis and Kurzyp-Wojnarska measured locus of control. These instruments measure generalized locus of control and allow the assessment of context-specific locus of control related to situations of success and failure, as well as school, parent, and peer settings. At the first stage of this study, students in non-integrated classrooms in schools without an inclusive curriculum were characterized by a more internal locus of control, both generalized and in situations of failure, compared to students of non-integrated classrooms in schools with an inclusive curriculum. At seventh grade, students of integrated classes were more external in situations related to their school activity, compared to their peers from non-integrated classrooms. Moreover, we observed developmental changes in locus of control of students from non-integrated classes but only those who attended schools with an integrated curriculum.

## 1. Introduction

Integrated education, which was introduced in Poland under the Education System Act of 7 September 1991 [1], allows typically developing and disabled students to participate in the general education curriculum together in one classroom. Integrated schooling embodies the idea of social integration through education. In contrast to segregated or special education, integrated education has been designed primarily to allow disabled children to attend schools near their place of residence and learn alongside their typically developing peers without being separated from their parents [2,3]. Today, after nearly 30 years of Polish experiences with integrated education, its strengths and weaknesses are apparent. The advantages of integrated schooling—as also confirmed by the experiences of countries other than Poland—include the following: students placed in integrated schools or classrooms learn responsibility, tolerance, and mutual acceptance. Students with disabilities are protected against social isolation [4,5] and are given a chance to be in a diverse environment. Integrated schools teach positive attitudes, mutual understanding, and ways of helping one another wisely [6]. While the advantages are clear, integrated education is not without its problems, such as the lower social position of disabled children in classes, limited social interaction between disabled students and their nondisabled peers, greater leniency of teachers, less demanding educational programs, and programs that are not adapted to the needs of disabled children [7].

An important measure that overcomes the disadvantages of integrated education is inclusive education, which has been implemented in many European countries [8]. Inclusion, to a much greater extent than integration, allows children with and without disabilities to fully co-participate in the educational process, ensuring mutual respect based on equal rights [9], and eliminates stigmatization of disabled students [10].

### 1.1. Education of Students with Disabilities and Their Nondisabled Peers

In Poland, the system of integrated education follows an integrated classroom. An integrated classroom accommodates disabled learners and learners without disabilities (there is a maximum of 20 students per class, including up to five students with a statement of special educational needs). Entering the period of adolescence together, they take on various biological, cognitive, and social challenges that naturally come with this period of life. Although the literature on how young people with various types of disabilities experience the period of adolescence is still scarce [11], the key characteristics of their functioning are readily outlined.

Disabled teenagers want to have the same sort of developmental experiences and opportunities as their typically developing peers, though they usually have to face more challenges because of their disability. Often, they cannot participate in school life and social life at the same level and pace as their peers without disabilities, which leads to negative psychosocial outcomes such as stress or loneliness [12,13]. They are also more frequently bullied by their peers [14].

Attitudes towards people with disabilities and towards their learning in integrated classrooms are often a source of misunderstanding and may act to the detriment of students who learn in integrated education settings. Depending on the nature of the disability, these students are perceived by others as being passive, non-self-reliant, dependent on others physically or mentally, and ill-adapted to the conditions in which they live [15]. The basic objective of integrated education is to create optimum conditions for these students, allowing them to achieve their educational and personal goals by using their own resources and potential.

Early adolescence is a time of formation of a young person’s character or personality. It can be assumed that the experiences of being placed in an integrated classroom can be a strong stimulus for developing openness and sensitivity to other people and their needs. Less numerous classes, a more diverse daily routine, the assistance of a support teacher, and the fact that the child stays in this supportive setting for several years are all conducive to their opening up to others and forming their own mature attitudes towards peers. The meeting of typically developing and disabled youth at school should therefore have a positive impact on both these groups of students, giving the former the possibility of interacting with their disabled classmates and opening up to their needs, and offering the latter the opportunity to interact with others in a natural social setting. As shown by Polish and international experiences with integrated education, these goals are not always achieved and so neither is real integration [16,17,18]. However, regardless of how successful integration is, placing disabled and nondisabled students in one classroom may affect the formation of their personality traits [19].

### 1.2. Research on Locus of Control in Adolescents with Disabilities

In Rotter’s theory of social learning, locus of control is understood as the way a person interprets the causes of various events that occur to them in their daily lives [20]. Events are viewed by people in terms of the degree they (events) can be controlled by them [21]. Rotter introduced this term to describe the way people perceive the relationship between their actions and the consequences of these actions. People with an internal locus of control believe they have control over events, both positive and negative, and attribute the outcomes of their actions to their own ability or lack thereof. They have a sense of control over the course of events in the world around them; they recognize that the outcomes of their actions are consistent with their behaviour and feel responsible for the decisions they make. In contrast, people with an external locus of control believe that things that happen to them in their lives are beyond their control and tend to attribute them to factors such as luck, chance, destiny, or unpredictable external circumstances [20,22].

With age and growing experience in distinguishing events that are causally related to one’s actions from those that are not, an external locus of control, which is characteristic of childhood, changes to a more internal locus of control, which increases in internality in adolescence. As people grow up, their competencies improve and so does their ability to influence events in their everyday lives. During this time, their locus of control undergoes intense changes under the influence of the educational environment of the family and school, in which the young person learns to take responsibility for their actions. In students’ daily lives, their locus of control allows them to decide what they want to learn and how they want to do it, and enables them to flexibly reconcile the need to fulfil their school requirements with the need to get engaged in out-of-school activities [23]. Therefore, it constitutes the core of the young person’s newly developing sense of responsibility, which in the future will become the basis for making personal choices.

Rotter and his concept is a personality-dominant approach to locus of control. Another approach, introduced in 1977 by Stern and Manifold, treats internal locus of control as a social value. The theory of the norm of internality stresses social judgements which favour internality as a way of interpreting different events [24,25]. Social superiority of internal persons springs from the fact that social appraisers, in schools or other institutions, prefer them more than externals. Norm of internality, as many research studies suggest, is particularly active in education as it plays an important role in assessment of children’s academic potential [26,27,28,29,30]. Dubois and Le Poultier [27] conducted studies in which teachers had to predict which fictitious students would pass into the next grade. They also obtained information about social class of pupil’s families (lower vs. higher) and a level of academic achievements of children (low vs. average). Children fulfilled an internality questionnaire (with external versus internal orientation), and teachers had to make predictions concerning their success in passing to another grade. Despite important cues concerning family status and academic potential, children who gave more internal explanations received more favourable judgments of the teachers. This result supports the notion about internality-based positivity bias which is maintained in situations involving information other than the evaluee’s internality or externality. Similarly, Bressoux and Pansu [31] found out that there were linear relationships between internality of pupils and positive judgements of their teachers. Therefore, the preference for internal causal explanations is learned at schools, while children discover that it is a socially desirable norm, which may bring benefits and, for this reason, might be deliberately chosen by them for self-presentation purposes [32].

Research indicates that the knowledge of students’ locus of control allows one to predict the quality of their adaptation and ability to cope with various challenges [33,34,35]. In particular, teachers working in integrated settings should be knowledgeable about their students’ locus of control to be able to plan teaching and educational activities for this specific group of learners. Locus of control, which is the focus of the analyses presented in this article, plays an important role in contemporary psychological discourse regarding the possibilities of predicting achievement, quality of school adaptation [36,37,38], and life satisfaction [39] in students placed in integrated and non-integrated programs, as well as the possibility of shaping future educational and professional careers of young people [40]. As research shows, people with an internal locus of control are more amenable to counselling and other forms of support than people with an external locus of control [41], which should be taken into account in planning teaching and educational activities in integrated settings. What is also important is that disabled youth with an internal locus of control have been observed to show significantly higher levels of disability acceptance compared to their peers with an external locus control [42]. Therefore, an internal locus of control can be viewed as a resource that allows people to achieve a better level of functioning. This is confirmed by studies of adolescents with physical disabilities, which show that people with an internal locus of control are more assertive and have better social skills [15].

Based on a literature review, Lawrence and Winschel [43] found that most four- and five-year-olds showed an external locus of control for both experiences of success and failure. With increasing age, people develop a more internal locus of control. In situations of success, most children aged 6 and 7 showed an internal locus of control, but it was only at the ages of 10–11 that young people started to attribute both success and failure to internal causes. Later studies, however, showed that children with learning disabilities developed a locus of control in a different way [44,45] and were characterized by a more external locus of control in both types of situations at a similar age, compared to children who did not have learning difficulties. Chapman and Boersma [46] noticed that children with disabilities developed a more internal locus of control with age, but only with regard to situations of failure.

A study of visually-impaired and hearing-impaired children showed that they had a more external locus of control [47,48,49] compared to children without such impairments. Janelle [15] compared the locus of control between physically disabled adolescent students and their typically developing peers. Her results indicated that there were no differences in the locus of control between students from these two groups. Moreover, studies by Center and Ward [50] on children with mild cerebral palsy who went to school together with their nondisabled peers showed that these children did not differ from their peers in terms of locus of control.

There is a noticeable lack of systematic research on the development of locus of control in students learning in integrated settings, a gap that the present study is intended to bridge. In these students, the locus of control may determine the way they interpret their school situation and affect the quality of their adaptation to the new environment and requirements. This applies to both disabled students and their typically developing classmates.

### 1.3. Aim of the Study

A review of previous studies on the locus of control in adolescents with disabilities and the assumptions of our own research project allowed us to formulate the following aims:To examine differences in the generalized and specific LOC related to success and failure in school, family, and peer situations between students in integrated and non-integrated programs; andTo investigate developmental changes in generalized and context-specific LOC related to success and failure in school, family, and peer situations in students from integrated and non-integrated programs over a three-year schooling period.

There are no contemporary systematic comparative studies on the specific character of the development of locus of control in students from integrated and non-integrated classes during early adolescence. We expect that the results we obtain will allow this gap in the literature to be filled and, in practical terms, will provide new conclusions and justification for further research in this area.

## 2. Materials and Methods

### 2.1. Participants

A total of 90 students attending integrated and non-integrated classes at three primary schools in Lublin, Poland, participated in the study. The study used a longitudinal design with three waves of data collected in the following years: the school year 2016/2017—stage I; the school year 2017/2018—stage II; and the school year 2018/2019—stage III. Once each school year, questionnaires were completed by the same students who, in the sequential research stages, were in grades 6, then 7, and then 8. The study was conducted in accordance with relevant ethical principles, with particular emphasis on confidentiality, anonymity, and voluntary participation. Consent from parents or caregivers was obtained for the students to participate in this study. In addition, permission to conduct the study was sought from headmasters of the schools concerned. All subjects gave their informed consent for inclusion before they participated in the study. The study was conducted in accordance with the Declaration of Helsinki, and the protocol was approved by the Ethics Committee for Scientific Research of Maria Curie-Skłodowska University (No. 29). Questionnaires were completed in groups during one meeting with each class. The participants were informed about the scientific purpose of the study and the anonymity of the results.

Students from seven classes, including three integrated classes, were surveyed. The students were divided into three same-size groups: students learning in integrated classes—30 individuals (50% girls) in schools with integrated classes; students attending non-integrated classes in schools with integrated classes—30 individuals (50% girls); and students from non-integrated classes in segregated school with a traditional general education curriculum—30 individuals (57% girls). In total, 11 students with disabilities learning in integrated classrooms took part in the study. The average ages of the participants in each stage of the study were as follows: stage I, *M* = 12.44, *SD* = 0.60; stage II, *M* = 13.63, *SD* = 0.52; and stage III, *M* = 14.51, *SD* = 0.69.

The respondents completed a personal data sheet and the Locus of Control Questionnaire (LCQ) by Krasowicz-Kupis and Kurzyp-Wojnarska [51,52]; in stage I of the study, they completed the original version of the LCQ, and in stages II and III, a revised version (LCQ-R). The questionnaires were administered during one 45-minute class period. As a token of appreciation for their participation in the study, each class received feedback via their tutor regarding their locus of control scores.

### 2.2. Measures

In stage I of the study (school year 2016/2017), locus of control was measured using LOQ (released in 1990), the Polish version of an instrument I-E, based on the original theory of social learning and locus of control of Rooter (1966), which was available at that time [51]. The difference between LOQ and I-E concerns the way of interpreting the results: in LOQ, the higher the score, the more internal a person is, and the reverse is true in I-E). LOQ was the only available tool in 2016/2017 to measure locus of control of adolescents. In later stages (II and III), a revised version of the instrument was used (LOQ-R), which was released in 2017 [52]. A new version of a tool was prepared in order to adjust items to the reality of the lives of contemporary adolescents. In particular, descriptions of situations were refreshed and broadened in several contexts: school, parents, and peers. Therefore, LOQ-R has more items than LOQ. Both versions of the LOQ are used to measure the participants’ sense of control over the outcomes of their behaviour, expressed on a continuum from an external to an internal locus of control.

The LOQ consists of 46 items, 36 of which have a diagnostic value. Half of the diagnostic items refer to situations of success and the other half to situations of failure; the sum of the scores on those items is a measure of a generalized locus of control, i.e., a general tendency to attribute more internal or more external causes to various events. The minimal raw score is 0, and the maximum is 18 for each scale: Success and Failure and 0–36 for a generalized locus of control. The questions are addressed directly to the respondent and ask about situations experienced by them. External factors responsible for success or failure are situations related to the family home, peers, and school. Internal factors, on the other hand, include a respondent’s own behaviours and efforts [51].Examples of Success item:If one day I manage to do everything that I have planned, it is because:(a)This is my lucky day(b)I try my best to do everything efficiently*Examples of Failure item*:If one day, you lost something valuable, it is because:(a)I had bad luck that day(b)I was careless

There are two versions of the LOQ sheet, one for girls and one for boys. They only differ in the grammatical form of the questions (Polish has gendered forms), the content of the questions is the same. The instrument has moderate psychometric properties. Cronbach’s *alpha* reliability coefficients for the individual scales are as follows: global scale *alpha* = 0.62; Success scale *alpha* = 0.40; and Failure scale *alpha* = 0.54. Despite the low values of the coefficients *alpha,* we decided to administer LOQ because at the moment of conducting stage I of this research, it was not possible to measure locus of control in adolescents with another test. The LOQ has a standard ten (sten) norms for people aged 13–17.

The LOQ-R sheet consists of 43 items, 38 of which have a diagnostic value. The test has only one form of answer sheet for both sexes, with each question containing two grammatical forms (feminine and masculine). Similarly to its predecessor, the LOQ-R allows one to calculate three basic types of score: a generalized locus of control (global) score and scores on the Success and Failure scales. Minimum raw score possible to obtain in Success scale is 0, maximum is 10; in Failure scale, it is 0–15, in a generalized locus of control from 0 to 38. Additionally, it can be used to measure locus of control in particular contexts: School, Parents, Peers, and Non-specific.Example of School item:If you participate in an important subject competition, it is because:(a)A teacher chose you(b)I decided to participateExample of Parents item:If your parents are angry about a mess in your room, it is because:(a)I am messy(b)They always find reasons to complainExample of Peers item:If your friend has a different opinion about something than yours, it means:(a)He will never change his mind, independently of what I do(b)I can influence his change of opinion

The LOQ-R has good psychometric properties. Reliability analyses have shown that the questionnaire has a high or satisfactory reliability. In the age range of 13–15 years, Cronbach’s reliability coefficient for the global score was *alpha* = 0.81; for the Success scale *alpha* = 0.63; and for the Failure scale *alpha* = 0.66. The instrument also has satisfactory theoretical validity. The LOQ-R has sten norms only for the Success, Failure and global scales. These norms cover the age range of 13–18 years.

High scores on each of the LOQ and LOQ-R scales are reflective of an internal locus of control, low scores point to an external locus of control, and medium scores indicate that the individual has not yet developed a locus of control. The global score is the sum of the scores obtained on the Success and Failure scales and is a measure of a person’s generalized locus of control. It provides information on a person’s generalized tendency to attribute responsibility for various events to internal or external factors. The subscale scores provide more specific information on how the respondent typically interprets the causes of events related to success and failure or the participant’s locus of control in different settings: school, parents, and peer situations.

## 3. Results

Data obtained through the LOQ and LOQ-R questionnaires regarding the Success scale, Failure scale, and the global scale were interpreted using sten norms appropriate for the respondents’ age. Because the specific scales of the LOQ-R, defining the respondents’ locus of control in school, parents, and peer situations, do not have sten norms, the analysis of that context-specific locus of control was based on raw scores. Statistical calculations were carried out using SPSS v.25. The results are analyzed in the order in which the research questions were asked at the beginning of the article. A one-way ANOVA and a Tukey post-hoc test were used to check what statistically significant differences there were in the locus of control scores between students from integrated vs. non-integrated classes at each stage of the study. To determine developmental changes in the locus of control characteristic of each group during the three-year study period, a repeated-measures ANOVA followed by a post-hoc test with the Bonferroni correction was performed. Prior to these analyses, the Mauchly test had been used to evaluate the assumption of data sphericity This test turned out to be statistically insignificant [53].

The results are shown in Table 1 and Table 2. Table 1 presents generalized locus of control scores and scores on the Success, Failure, School, Parents, and Peers scales obtained by students from integrated classes (group 1) and non-integrated classes (group 2). Table 2 compares generalized locus of control scores and scores on the Success, Failure, School, Parents, and Peers scales obtained by students from integrated classes and non-integrated classes in the three successive stages of the study (grades 6, 7, and 8). These scores were used to evaluate the developmental changes in the locus of control that occurred in the students from the three investigated groups over the study period.

The one-way ANOVA identified differences between groups in the generalized locus of control, and in locus of control in situations of success and failure as well as situations occurring in school, family, and peer settings for each research stage. Statistically significant effects were found for the generalized locus of control (F (2,87) = 2.98; *p* < 0.05) and locus of control in failure situations (F (2,87) = 2.48; *p* < 0.05) in research stage I and for locus of control in school situations (F (2,87) = 2.48; *p* < 0.05) in stage III of the study. Post-hoc comparisons using the Tukey test revealed the following statistically significant differences between groups: In the first stage of the study (grade 6), the generalized locus of control and locus of control in failure situations statistically significantly (*p* < 0.05) differentiated students from non-integrated classes. Students who went to non-integrated classes in schools without an inclusive curriculum were characterized by a more internal locus of control, both generalized and in situations of failure. Post-hoc test data for stage III of the study (grade 7) revealed statistically significant differences (*p* < 0.05) in locus of control in the school setting between students from integrated and non-integrated classes. Students who learned in integrated classrooms showed a more external locus of control in situations related to their school activity compared to their peers from non-integrated classrooms.

To determine developmental changes occurring in each of the examined groups during the three-year measurement of locus of control, a repeated-measures ANOVA was performed. This analysis revealed significant main effects for locus of control in non-integrated classes only. For students from non-integrated classes who attended a school with an integrated program, a significant main effect was found for locus of control in situations of failure (F (2,27) = 4.64; *p* < 0.05; η^2^ = 0.14) and generalized locus of control (F (2,27) = 3.32; *p* < 0.05; η^2^ = 0.10). The effect for locus of control in failure situations was strong and explained 14% of the variance in this context-specific locus of control over the three-year study period. In the case of the generalized locus of control, the effect was moderately strong and explained only 10% of the variance. For students from the school with a traditional teaching program, a statistically significant main effect was found for locus of control in situations of success (F (2,27) = 4.58; *p* < 0.001; η^2^ = 0.02). This effect was moderately strong and explained 2% of the variance.

The post-hoc test with the Bonferroni correction showed that in the case of students from non-integrated classes who attended a school with integrated classes, there were significant differences in locus of control in failure situations and generalized locus of control between the first and second measurements (grades 6 and 7). Both the students’ locus of control in situations which they saw as a failure and their generalized locus of control were statistically significantly stronger in the second measurement than in the first one. Accordingly, these students had a significantly more internal generalized locus of control (*p* < 0.05) and a more internal locus of control in situations of failure (*p* < 0.05). Additionally, the post-hoc test showed that there were statistically significant differences between the first and second measurements in the success-related locus of control in students from the non-integrated school. Their locus of control in situations of success was significantly weaker in the second measurement (grade 7) compared to the first measurement (grade 6). This means that these students’ locus of control in situations of success had shifted significantly (*p* < 0.01) towards externality.

## 4. Discussion

The present study explored developmental changes in the locus of control in students placed in integrated classes and their peers from non-integrated classes right after the 2016 reform of the Polish education system [54]. The locus of control of students, especially those attending integrated classes, is rarely investigated, as evidenced by the lack of contemporary research in this area in Polish and world literature. In this study, we compared the developmental changes in the locus of control in young people learning in different educational contexts: integrated and non-integrated classes during a three-year schooling period. The participants were the first generation of students taught in the new system refashioned by the reform, which liquidated middle schools and extended primary education from six to eight years. This study is original because it was carried out in this unique context and compares the results obtained by students over a three-year period.

The development of locus of control during early adolescence is affected by various factors. One of them is the pubertal growth spurt which modifies adolescents’ self-esteem and emotions, leading to changes in the way they interpret various events [55]. Other factors are changing external circumstances, such as transition to the next educational stage, and the need to adapt to the new requirements and the new peer environment [22,23,56,57]. Maxey and Beckert [11] stress the importance of relationships with parents, siblings, and friends; the impact of school; access to new technologies; and involvement in extracurricular activities, as factors that play a role in the psychosocial functioning of adolescents and affect their locus of control.

The first aim examined differences in the generalized locus of control and context-specific locus of control associated with situations of success and failure, and school, parents and peer settings, between students attending integrated and non-integrated classes in stages I, II, and III of the study. In the first stage of the study (grade 6), the generalized locus of control and locus of control in situations of failure were statistically significant for students from non-integrated classes. Students who learned in non-integrated classrooms in schools without an inclusive curriculum were characterized by a more internal locus of control, both generalized and associated with situations of failure, compared to their peers from non-integrated classes in integrated schools. This result can be explained by the fact that in schools with integrated classrooms, students who are not in an integrated program are taught in a similar way to students from integrated classes. It is not uncommon for teachers in these schools to teach in both general education and integrated classrooms, which means they may set the same requirements and use the same educational methods in both types of classrooms, as part of their established ways of working with young people. The shift towards an external locus of control in situations of failure observed in the students from integrated schools may be the result of their receiving greater support and help from teachers in difficult and frustrating situations. In the case of schools without integrated classrooms, the requirements and teaching programs are the same for all classes. In settings in which all classes are “the same” and all students have equal intellectual and physical capacities, they are all treated in the same way, receive the same range of support, and have to fulfil the same requirements. This may explain why the students in traditional general education classes had a more internal locus of control than their peers from non-integrated classes who attended integrated schools.

The second aim was related to the character of developmental changes in generalized locus of control, locus of control in situations of success and failure, and locus of control in school, family, and peer settings in students learning in integrated and non-integrated classrooms. The analyses showed that between the first and second stage of the study, developmental changes occurred in the locus of control of the participants from non-integrated classes, but only those who went to schools with an integrated curriculum. There were significant differences in these students’ locus of control in failure situations and their generalized locus of control between stage I (grade 6) and stage II of the study (grade 7). The students’ locus of control in situations of failure and their generalized locus of control were both statistically significantly stronger in the second measurement than in the first one, which meant that there had been a shift towards internality. This result is interesting and may indicate that—despite the fact that a year earlier, these students had been more likely than their peers from schools with traditional curriculum to attribute the causes of various events, including failures, to external factors—after a year, they began to show greater responsibility for their actions and choices. Perhaps, at that time (stage II, spring 2017), the students had already adapted to their new educational situation and felt more confident; at the same time, teachers began to be more demanding towards them, which resulted in a shift towards a more internal locus of control. This shift may have also been a natural effect of the developmental changes that people typically undergo in early adolescence [22].

Even more intriguing is the result regarding developmental changes in the success-related locus of control in students from schools with a traditional curriculum. They had a significantly weaker locus of control in situations of success in the second measurement (grade 7) than in the first measurement (grade 6). It is worth recalling that in the first measurement, these students were characterized by a significantly stronger internal generalized locus of control and a stronger locus of control in situations of failure compared to their peers from non-integrated classes who went to integrated schools. Perhaps they focused on being responsible for their personal failures so much that they felt much less convinced that their successes were also an effect of their own activity. It is also possible that the weakening of locus of control in situations of success was a consequence of the fact that the level of work became more demanding in the seventh grade, and the youth experienced stress associated with being the first year of students continuing their education in a primary school instead of transitioning into a middle school [58].

In stage III of the study, statistically significant differences in the locus of control were observed between students from integrated and non-integrated classes. They concerned the locus of control in school situations. Students who pursued an integrated program had a more external locus of control observed in situations related to their school activity compared to their peers from classes with traditional instruction. This means that the students from integrated classes were more likely to attribute their academic achievement to external factors instead of seeing it as an outcome of their own actions and efforts. This result is worrying, as at the time of the third stage of the study, these students were facing the prospect of choosing a secondary school and meeting the criteria set for all candidates. A lack of a sense of control over school situations, such as tests or exams, could reduce their chances of being accepted into the school of their choice. This result may also be indicative of the students’ uncertainty about their further educational career. The fact that not all secondary schools in Lublin have inclusive programs may have reduced the motivation of the students with disabilities to actualize their potential and pursue their dreams of continuing education in a school they wanted to attend. This, in turn, may have led to a feeling of alienation and uncertainty about their future. The finding that students from integrated classes have a more external locus of control is consistent with the observation made by Clark et al. [59] that there exist threats connected with integrated education that may make it difficult for this group of children to develop an internal locus of control in the school setting. They include, among others, excessive reliance of the students on the teacher and a limited possibility of making independent choices.

One of the tasks of integrated education is to help eliminate differences in the development of locus of control between children with special educational needs and children without developmental disorders studying in one class [45]. Our research showed that the changes in the educational system triggered by the structural reform of 2016 did not amend the psychological situation of students in integrated classes. Their locus of control in situations of success and failure was still shifted towards externality. As already signalled in the literature, it is related to society’s belief that people with disabilities have a lowered self-control and must not only be assisted in their daily activities but also must have their decisions made for them [60]. It is also worth paying attention to the fact that the psychological situation of teachers who work in integrated schools differs from that of their colleagues who teach in traditional classes. Visser [61] mentions that the former are under a lot of mental strain and suffer increased levels of stress as they often experience strong and extreme emotions or have difficulty coping with frustration, which may incline them to allow more lenient conditions for their students.

Some authors [62,63,64] point out that the development of locus of control is influenced by individual life experiences, including those related to schooling. By finding out how young people interpret the causes of various events, one can identify their current needs and aspirations and understand how they perceive themselves in the world in which they live. Thus, the present study makes a contribution to methods of working with students with and without special educational needs [5,18]. From a practical point of view, it provides an important insight that the development of an internal locus of control in the school setting is encouraged by teacher attitudes that promote autonomy in making decisions, including those related to learning, and encourage students to be proactive. Teachers who motivate their students to show initiative and try their hand at various experiences help them build a realistic picture of themselves. Regardless of whether it is in an integrated or a non-integrated classroom, students who are successful in their efforts develop a positive self-image and self-confidence.

## 5. Conclusions

Despite the fact that this study was conducted in a relatively short longitudinal period, it was enough to determine changes in locus of control of students. The results indicate differences of locus of control development of students attending different types of classrooms and schools. In general, the results indicate the role of educational context on the development of locus of control in adolescents with and without disabilities. Among the complex underpinnings of educational influences on personality development in adolescence in the case of integrated education, the most important are a system of rewards and motivators and psychological preparation of teachers who work in this type of education.

The conclusions of this study, though interesting, need to be interpreted with caution. Results, especially those obtained in the first stage of this study, are burdened with inaccuracy due to a low internal consistency of the LOQ [65]. There are different reports about the acceptable values of *alpha* coefficients in psychological tests, ranging from 0.70 to 0.95 [66,67]. Instruments which have *alpha* below 0.60 (as it is in the LOQ), due to their low reliability, theoretically should not be used in the research. In spite of this, we decided to administer LOQ because, at the moment of conducting stage I of this research, it was not possible to measure locus of control in adolescents with another test. Another limitation of this study is the study cohort. It was relatively small, with 30 students from each of the analyzed groups surveyed. Moreover, the study cannot be replicated in the future because the context of the study has already changed: the new reform, which was being introduced at the time of the experiments, now is fully implemented. It is still worth further exploration of the developmental changes in the locus of control among students from integrated and non-integrated classes in the new educational reality.

## Figures and Tables

**Table 1 behavsci-10-00074-t001:** One-way ANOVA of differences between locus of control scores obtained by students from integrated and non-integrated classes during the three stages of the study: in the school years 2016/2017 (research stage I), 2017/2018 (research stage II), and 2018/2019 (research stage III).

Research Stage	Locus of Control	Type of Class			ANOVA		Tukey Post Hoc Test *p* <		
		1	2	3					
		M (SD)	M (SD)	M (SD)	F	*p*	1–2	1–3	2–3
I	Success	4.97 (1.79)	4.83 (2.17)	5.90 (2.09)	2.48	0.090	n.s.	n.s.	n.s.
	Failure	4.23 (2.45)	3.20 (1.99)	4.50 (2.15)	2.92	**0.044 ***	n.s.	n.s.	**0.041 ***
	Generalized	4.20 (2.47)	3.63 (2.02)	5.03 (2.19)	2.98	**0.046 ***	n.s.	n.s.	**0.045 ***
II	Success	4.57 (1.76)	5.33 (2.04)	4.43 (1.77)	2.04	0.136	n.s.	n.s.	n.s.
	Failure	3.60 (2.31)	4.70 (2.25)	4.37 (2.27)	1.84	0.164	n.s.	n.s.	n.s.
	Generalized	3.73 (2.16)	4.80 (2.17)	4.53 (2.10)	2.01	0.140	n.s.	n.s.	n.s.
	School	7.93 (3.23)	8.33 (2.40)	9.03 (2.48)	1.25	0.293	n.s.	n.s.	n.s.
	Parents	5.77 (1.97)	6.87 (1.71)	6.60 (2.31)	2.43	0.094	n.s.	n.s.	n.s.
	Peers	4.63 (1.65)	5.33 (2.05)	4.60 (1.94)	1.44	0.243	n.s.	n.s.	n.s.
III	Success	4.27 (2.16)	4.95 (2.08)	4.70 (2.43)	0.69	0.505	n.s.	n.s.	n.s.
	Failure	3.70 (2.13)	4.27 (2.16)	4.13 (2.13)	0.57	0.566	n.s.	n.s.	n.s.
	Generalized	3.73 (1.93)	4.37 (1.85)	4.50 (2.42)	1.16	0.317	n.s.	n.s.	n.s.
	School	7.53 (2.57)	8.93 (2.35)	8.97 (2.77)	3.04	**0.043 ***	**0.044 ***	**0.044 ***	n.s.
	Parents	5.53 (2.13)	6.23 (1.48)	6.23 (2.11)	1.32	0.274	n.s.	n.s.	n.s.
	Peers	4.97 (1.54)	4.63 (1.25)	4.77 (1.85)	0.34	0.710	n.s.	n.s.	n.s.

Note: * *p* < 0.05. Legend: 1—integrated class (school with integrated classes); 2—non-integrated class (school with integrated classes); 3—non-integrated class (school without integrated classes); n.s.—not statistically significant.

**Table 2 behavsci-10-00074-t002:** Repeated-measures ANOVA of locus of control scores obtained by students from integrated and non-integrated classes in the school years 2016/2017 (research stage I), 2017/2018 (research stage II), and 2018/2019 (research stage III).

Type of Class	Locus of Control	Research Stage			ANOVA		Bonferroni Post Hoc Test; *p* <		
		I	II	III					
		M (SD)	M (SD)	M (SD)	F	*p*	I–II	I–III	II–III
1	Success	4.97 (1.79)	4.57 (1.76)	4.27 (2.16)	0.94	0.398	n.s.	n.s.	n.s.
	Failure	4.23 (2.45)	3.60 (2.31)	3.70 (2.13)	0.70	0.503	n.s.	n.s.	n.s.
	Generalized	4.20 (2.47)	3.73 (2.16)	3.73 (1.93)	0.41	0.668	n.s.	n.s.	n.s.
	School	-	7.93 (3.23)	7.53 (2.57)	0.46	0.503	-	n.s.	n.s.
	Parents	-	5.77 (1.97)	5.53 (2.13)	0.15	0.702	-	n.s.	n.s.
	Peers	-	4.63 (1.65)	4.97 (1.54)	0.58	0.452	-	n.s.	n.s.
2	Success	4.83 (2.17)	5.33 (2.04)	4.95 (2.08)	0.59	0.559	n.s.	n.s.	n.s.
	Failure	3.20 (1.99)	4.70 (2.25)	4.27 (2.16)	4.64	**0.014 ***	**0.042 ***	n.s.	n.s.
	Generalized	3.63 (2.02)	4.80 (2.17)	4.37 (1.85)	3.32	**0.043 ***	**0.047 ***	n.s.	n.s.
	School	-	8.33 (2.40)	8.93 (2.35)	0.49	0.553	-	n.s.	n.s.
	Parents	-	6.87 (1.71)	6.23 (1.48)	0.39	0.543	-	n.s.	n.s.
	Peers	-	5.33 (2.05)	4.63 (1.25)	3.32	0.079	-	n.s.	n.s.
3	Success	5.90 (2.09)	4.43 (1.77)	4.70 (2.43)	4.58	**0.014 ***	**0.003 ****	n.s.	n.s.
	Failure	4.50 (2.15)	4.37 (2.27)	4.13 (2.13)	0.23	0.793	n.s.	n.s.	n.s.
	Generalized	5.03 (2.19)	4.53 (2.10)	4.50 (2.42)	0.58	0.562	n.s.	n.s.	n.s.
	School	-	9.03 (2.48)	8.97 (2.77)	0.01	0.931	-	n.s.	n.s.
	Parents	-	6.60 (2.31)	6.23 (2.11)	0.38	0.541	-	n.s.	n.s.
	Peers	-	4.60 (1.94)	4.77 (1.85)	0.12	0.731	-	n.s.	n.s.

Note: * *p* < 0.05. ** *p* < 0.01. Legend: 1—integrated class (school with integrated classes); 2—non-integrated class (school with integrated classes); 3—non-integrated class (school without integrated classes); n.s.—not statistically significant.

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
