# Peer review of "Developmental Changes in the Locus of Control in Students Attending Integrated and Non-integrated Classes during Early Adolescence in Poland"

_behavsci, 2020, doi:10.3390/bs10040074_

Round 1

Reviewer 1 Report

Thank you for this is an interesting paper which contributes to the knowledge on developmental Locus of control (LOC) .

There are however some corrections that are needed.

My comments are as follows:

Line 25 should say –measures

Line 37 – the advantages not the upsides and that sentence is too long.

Line 53 …system of education follows an integrated….

Line 58 …the key characteristics…

Line 72  of a young person’s character or personality… rather than various traits

Line 146  Aim is different from the research questions and maybe this sections should be labelled research questions??? Under the existing title the research questions need to be revised. Irrespective of this the questions are too convoluted and should be tighter. Suggestions offered below;

Question 1

To examine differences in the generalised and specific LOC related to success, and failure in school, family and peer situations between students in integrated and non-integrated programs.  

Question 2

To investigate developmental changes in generalised and context specific LOC related to success and failure in school, family and peer situations in students from integrated and non-integrated programs over a three year schooling period.  

Line 195 I think a sentence is needed on how this questionnaire is based on the original generalised LOC (Rotter, 1966). There are more items than in the Rotter’s version and the scoring is in contrast to the original LOC, where low scores indicated internality and high scores indicated externality. A reader who is conversant with the original LOC will find this confusing.

Line 207 I think an explanation is needed why revised version is used and why it has different number of items.

Line 242 explanation for group 3 needs to be added to the sentence.

Line 250 add to legend - * probability .05

Line 256 where is these probabilities shown, maybe highlight on the table

Line 308 remove novelty. This study is original because it was carried out…..

Line 318 The first research question examined differences….

Line 322 ….in situations of failure was statistically significant for students….

Line 333 unless I am reading it wrong, you contradict your findings in the sentence. Young people from non-integrated classes….

Line 378 This result is worrying as at the time…

Line 397 ….have a lower self-control…

Line 418  …30 students…

Line 419 the study cannot be replicated in the future…

Line 421 now fully implemented it is still worth further exploration of the developmental changes….

Author Response

Thank you for very detailed and helpful review of article.

Thank you for very accurate indications for improving the English style of expression in text.

 All your tips have been included in the article (lines 25,37,53,58,72, 250, 256, 308, 318, 322, 378, 397, 418, 419, 421).

In addition, the research questions were reformulated according to the reviewer's suggestion (line 146).

With regard to the remark about the need to explain why two methods of testing the control location were used and how they relate to the original method (lines 195 and 207) we make the appropriate addition in the text:  We broadened descriptions of LOQ and LOQ-R.

With regard to the remark about the need to describe group 3 (line 242), we have supplemented the relevant part of the article: We supplemented a necessary sentence.

Referring to the note noted on line 333: We admit this sentence is misleading.

Reviewer 2 Report

This study presents a important public health topic. The manuscript is well-written. Great job!

I have only one comment on the presentation of Table 1: What are those scores means? Can you cite a questionnaire or put the questionnaire in the supplementary document?

Author Response

Thank you for preparing a review of article.
Our response to the comments contained in the review:

Minimum score which is possible to obtain in LOQ  is 0, maximum is 36  and LOQ-R is 0 maximum is 38. The higher score, the more internal locus of control is.

We illustrated questionnaires we used by examples of items from each scales: failure, success, school, peers and so on.

Reviewer 3 Report

The article presents a longitudinal study on the development of a context of locus of control linked to success and failure situations in Polish adolescents.
LOC is therefore a general expectation of control, internal vs external, of the reinforcements of behavior produced by a person and, as such, it has been and is still often considered as a general personality variable.
A great deal of empirical research has made it possible to show more and more that internality is the subject of a social norm of judgment called: the norm of internality. The article does not present any research from this perspective and I consider that a great theoretical lack.
The bibliography used to describe LOC is not recent ...It would be useful to give some examples on the questionnaires used to be able to appreciate the analyzes carried out. Several questionnaires are marked by methodological biases depriving the data of their validity (Gangloff, 1997).

Author Response

Thank you for preparing a review of  article.

Our response to the comments contained in the review:

  1. In the theoretical part of article, we supplemented theoretical revisions on locus of control on the norm of internality concept, as an alternative for Rotter’s original theory of locus of control.
  2. We added more recent bibliography related to locus of control and theory of norm of internality
  3. We illustrated questionnaires we used by examples of items from each scales: failure, success, school, peers and so on.

Reviewer 4 Report

Your study is interesting and I would like to make the following recommendations:
I recommend that in the abstract be mentioned the most relevant conclusion.
I recommend that the conclusions be redone based on the results.

Author Response

Thank you for preparing a review of article.
Our response to the comments contained in the review:

  1. We put the most important results from the study in abstract section
  2. We redone conclusion sections with the most important results

Round 2

Reviewer 3 Report

Dear authors,

The improvements to the work are to be appreciated. However the article does not present any research from the perspective of normativity of internality.

Author Response

According to the Reviewer's suggestion, the theoretical part of the article has been broadened with both general assumptions of the theory of the norm of internality and conducted research in this field in the educational environment. The research  was presented in the separate paragraph  as constitute an important input to the analysis of developmental changes of locus of control in integrated classes.